# 1 Interplay between aerosol and updraft velocity in Large Eddy

# **2** Simulations of marine stratocumulus clouds

- Gaurav Dogra<sup>1</sup>, Olivier Boucher<sup>1</sup>, Nicolas Bellouin<sup>1,2</sup>
- <sup>1</sup> Institut Pierre-Simon Laplace, Sorbonne Université / CNRS, Paris, France
- <sup>2</sup> Department of Meteorology, University of Reading, Reading, UK
- Correspondence to: gaurav.dogra@ipsl.fr
- **Abstract:** Marine stratocumulus are low-level clouds with a great impact on the Earth's energy balance.
- The present study is focused on understanding the interplay between aerosols and updraft velocity in
- marine stratocumulus clouds using Large Eddy Simulations (LES) over a 6.4×6.4 km² domain size with
- a double-moment aerosol-cloud microphysics scheme. A first series of experiments with aerosol
- concentrations varying from pristine to polluted conditions shows a transition from aerosol-limited to
- updraft-limited regime. The higher aerosol concentration in polluted conditions leads to the suppression
- of precipitation due to a larger number of cloud droplets, suggesting a transition from an open-cell to a
- closed-cell structure. A second series of experiments, where updraft velocity is enhanced by increasing
- latent heat flux, shows an increase in vertical velocity variance and a higher cloud droplet number,
- indicating enhanced convective activity with stronger updrafts and downdrafts. Cloud susceptibility is
- equal to 1 for both experiments at lower aerosol concentration, clearly indicating the presence of an
- aerosol-limited regime where updraft velocity has little impact. At higher aerosol concentration, cloud
- susceptibility is higher for stronger updrafts in the second series of experiments, indicating that stronger
- updrafts can shift regime from updraft-limited to aerosol-limited. Stronger updrafts also influence aerosol
- availability and activation, blurring the distinction between aerosol-limited and updraft-limited regimes
- because of the key role updraft velocity plays in regulating aerosol activation. Overall, the study
- demonstrates that LES is capable of reproducing both regimes as well as the transition between them.

### 1. Introduction

- Marine stratocumulus clouds are widespread over the ocean and impact the Earth's energy budget. Being
- low-level clouds, their effect on longwave radiation at the top of the atmosphere is minimal due to their
- limited vertical extent. However, they reflect shortwave radiation back to space and produce a net top of
- the atmosphere cooling effect, with estimates ranging from -11.5 W/m<sup>2</sup> (Chen et al., 2000) to -8.2 W/m<sup>2</sup>
- (L'Ecuyer et al., 2019).
- Marine stratocumulus clouds are warm clouds made up of liquid droplets that nucleate from aerosols
- serving as cloud condensation nuclei (CCN), with droplet activation occurring at typical supersaturation
- levels in the range 0.1 to 2% (Yau and Rogers, 1996; Pruppacher and Klett, 2012). Thus, any change in
- aerosol amounts or properties will readily impact stratocumulus microphysical and radiative properties,

with further impact on the cloud evolution, including updraft velocities, precipitation formation, and cloud 35 lifetime (Albrecht, 1989). Despite advances in understanding and measuring aerosol-cloud interactions, 36 considerable uncertainties persist in contemporary global climate models, hampering the quantification 37 38 of aerosol radiative forcing (Stier et al., 2013; Bellouin et al., 2020; Quaas et al., 2020; Forster et al., 2021) with knock-on effects on climate projections (Fyfe et al., 2021; Michibata et al., 2025; Watson-39 40 Parris and Smith, 2022). 41 Twomey (1959, 1974) hypothesized that an increase in aerosol concentration leads to an increase in cloud droplet number concentration when cloud liquid water remains fixed, an effect now known as the 42 Twomey effect. This increase in droplet number results in smaller droplet sizes, which increases the 43 cloud's reflectivity to shortwave radiation, so that interactions of anthropogenic emissions of aerosols 44 with clouds are expected to cause a cooling effect on Earth. The cloud susceptibility  $\beta = \frac{\partial \ln N_d}{\partial \ln N_o}$ , is defined 45 as a measure for the cloud droplet number (N<sub>d</sub>) dependency on aerosol number (N<sub>a</sub>). However, aerosol 46 47 activation into a cloud droplet requires that ambient water vapor supersaturation exceeds the aerosol's critical supersaturation, which depends on its size and chemical composition (Köhler, 1921). 48 49 Supersaturation is typically achieved when the air parcel is lifted adiabatically, leading to expansion, 50 cooling, and an increase in relative humidity due to decreased saturation vapour pressure with decreased temperature. Thus, the aerosol takes up water by condensational growth and is a sink for relative humidity. 51 52 However, the smallest aerosol size that can be activated depends upon maximum supersaturation, which is further determined by the interplay between the increase in relative humidity due to lifting and the loss 53 54 of vapor by condensational growth (Lohmann et al., 2016). Thus, for a fixed aerosol population, activation depends upon the air parcel's updraft velocity (Pruppacher and Klett, 2012). 55 56 Reutter et al. (2009) showed with a parcel model that the interplay of aerosol concentration and updraft 57 velocity can be categorised in three distinct regimes. The first one is the aerosol-limited regime, corresponding to low aerosol and/or high updrafts. In this regime, the updrafts are strong enough to 58 activate a large fraction of the aerosols, due to the large supersaturation that can be generated by these 59 strong updrafts, leading to  $\beta \approx 1$ . The second one is the updraft-limited regime, corresponding to high 60 61 aerosol and/or low updrafts, in which updrafts limit aerosol activation. In this regime, an increase in aerosol does not increase N<sub>d</sub> much, leading to smaller values of β, and only an increase in updrafts can 62 increase cloud droplet numbers further. The third regime is sensitive to both aerosol number and updrafts, 63 with β in the range from 0 to 1. It corresponds to the transition between the updraft-limited and aerosol-64 65 limited regimes. These regimes have also been identified in many observational studies (Hudson and 66 Noble, 2014; Bougiatioti et al., 2016, 2020; Georgakaki et al., 2021; Guy et al., 2021; Kacarab et al., 67 2020; Misumi et al., 2022), and the importance of accurately simulating these regimes from a global modelling perspective is also highlighted by Sullivan et al. (2016). These regimes strongly influence 68

aerosol-cloud interactions; however, substantial uncertainties remain to correctly simulate these regimes
in global, coarse-resolution climate models.

Large-eddy simulations (LES), along with cloud-resolving models, are used extensively to investigate cloud-scale processes (Stoll et al., 2020; Dogra et al., 2025) and have emerged as valuable tools for advancing our understanding of aerosol–cloud interactions (Prabhakaran et al., 2024; Schwarz et al., 2024). They also play a crucial role in enhancing the predictive capabilities of larger-scale atmospheric models (Grabowski et al., 2019). In a recent study, Schwarz et al. (2024) evaluated the ability of LES to reproduce aerosol cloud regimes in marine stratocumulus environments using a bulk microphysics scheme (Seifert and Beheng, 2001, 2006). Their default model configuration, which employed a coarse time step of 1 second and a relatively small initial radius for newly activated droplets, was unable to capture the updraft-limited regime. However, they successfully achieved regime transitions by reducing the model time step to approximately 0.1 seconds and either applying renormalization to the activated droplet population, such that the water mass of the newly activated droplet does not exceed that allowed by local supersaturation, or by increasing the initial droplet radius. Motivated by these findings, we aim to investigate the transition of aerosol-updraft limited regimes in a marine stratocumulus cloud using LES coupled with a two-moment CASIM (Field et al. 2023) microphysics scheme, which includes an explicit representation of aerosol–cloud interactions.

# 2. Methodology

MONC includes prognostic momentum, energy, and moisture equations and parametrizations for subgrid-scale turbulence and radiation. It is coupled with the CASIM (Cloud AeroSol Interacting Microphysics) scheme (Field et al., 2023), i.e., a double-moment microphysics scheme that actively tracks both the mass and number concentration of hydrometeors (cloud water, rain, ice, etc.) and of aerosols. It represents multiple aerosol modes – Aitken, accumulation, and coarse – and allows for aerosol activation into cloud droplets based on local supersaturation (Abdul-Razzak and Ghan, 2000), enabling a physically based

Simulations are conducted using the Met Office NERC Cloud Model (MONC; Brown et al., 2020).

representation of aerosol-cloud interactions. CASIM also includes parameterizations for collision

coalescence, auto conversion, and sedimentation, making it suitable for detailed microphysical studies.

The simulations are initialized using observational data from the DYCOMS-II field campaign (Stevens et al., 2003), which provides initial and boundary conditions for night-time stratocumulus clouds and is also a well-established intercomparison study for stratocumulus clouds (Ackerman et al., 2009). The simulations are initialised with vertical profiles of potential temperature, specific humidity, u and v winds. A domain-wide divergence of  $3.75 \times 10^{-6} \, \mathrm{s}^{-1}$  is imposed as large-scale vertical velocity forcing. The model domain consists of  $128 \times 128$  grid points in the x, and y directions, with horizontal grid spacing of 50 m, resulting in a horizontal domain of  $6.4 \times 6.4 \, \mathrm{km}^2$ . There are 150 grid points in the vertical, where

the grid is stretched, with finer resolution (~7 m) near the cloud top and coarser resolution (~40 m) in

the free troposphere. Periodic boundary conditions are used for lateral boundaries, and rigid lids are used 104 105 for both top and surface boundary conditions. Newtonian damping is applied above 1250 m to top lid in order to prevent gravity wave perturbations. Subgrid turbulence is parametrized using the Lilly (1962) 106 107 and Smagorinsky (1963) models. For radiation, the Suite of Community Radiative Transfer codes based on Edwards and Slingo (SOCRATES) (Edwards and Slingo, 1996) is used. 108 109 Simulations are run for 6 hours, with a dynamic timestep of 0.4 seconds. Surface fluxes are constant at default values of 16 W/m<sup>2</sup> (sensible heat) and 96 W/m<sup>2</sup> (latent heat), giving a Bowen ratio (ratio of 110 sensible to latent heat flux) of approximately 0.17. The output is saved every 300 seconds to analyse the 111 results. 112 The study is divided into two types of experiments. The first type is denoted AERO. It is used to study 113 114 the transition to an updraft-limited environment by varying the aerosol number concentration from pristine to heavily polluted conditions. AERO consists of 5 simulations where the initial aerosol number 115 concentration (Na) is varied across four cases: 65, 100, 500, 1000, 10000 cm<sup>-3</sup>, assuming in all cases an 116 accumulation-mode lognormal size distribution with mode radius = 0.06 µm and geometric standard 117 deviation  $\sigma = 1.7$  following Schwarz et al. (2024). The AERO cases are abbreviated as A-65, A-100, A-118 119 500, A-1000, and A-10000. The second type of experiments, denoted BRATIO, is performed to study the shift from updraft-limited 120 to aerosol-limited conditions by repeating the AERO experiments with a Bowen ratio decreased to 0.06 121 122 by keeping the sensible heat flux constant, which results in a latent heat flux value of 266 W/m<sup>2</sup>. The 123 results from BRATIO experiments, with their corresponding aerosol number concentrations, are abbreviated as BR-65, BR-100, BR-500, BR-1000, and BR-10000. 124

125126

131132

133134

## 3. Results

# 128 3.1 Updraft-limited regime

We first analyse the AERO simulations to study the transition to an updraft-limited regime in marine stratocumulus clouds with increasing aerosol number from 65 cm<sup>-3</sup> to 10<sup>4</sup> cm<sup>-3</sup>. Figure 1 shows a comparison of the horizontally averaged time evolution of the liquid water path (LWP), cloud base, cloud height, and precipitation rate across the experiments. The first two hours of simulations in semi-transparent grey box are considered as spin-up, as the dynamical features are sensitive to the initial forcing.

**Figure 1** Horizontally averaged time series of a) Liquid water path (g m<sup>-2</sup>), b) cloud base height (m), c) cloud top height (m), and d) precipitation rates (mm d<sup>-1</sup>) for AERO simulations with aerosol concentrations of 65 (teal green, A-65), 100 (orange, A-100), 500 (purple, A-500), 1000 (magenta, A-1000), and 10000 (sky blue, A-10000) cm<sup>-3</sup>.

The impact of increasing aerosol concentration on LWP is a two-way effect. The first increase in aerosol concentration from 65 to 100 cm<sup>-3</sup> translates into an increase in LWP, due to a larger number of cloud droplets with smaller size formed in A-100 compared to A-65. This leads to a reduction in droplet removal by sedimentation and leads to a decrease in precipitation rate as observed in Fig. 1d. However, upon further increase in aerosol concentration, in A-500, A-1000 and A-10000, a decrease in LWP is observed, which is consistent with past studies (Hoffmann and Feingold, 2019; Xue and Feingold, 2006). This is mainly due to the entrainment-evaporative feedback, as the cloud droplets formed are smaller and more prone to evaporation, thus leading to a stronger entrainment rate, as shown in Appendix A. Moreover, a

heightened cloud base and top are also observed with increasing aerosol concentration, which can be attributed to precipitation suppression.

Snapshots of the liquid water path (LWP) distribution at the end of the third hour for all AERO simulations are shown in the top row of Fig. 2. The cloud structure exhibits significant changes with varying aerosol concentrations. These snapshots indicate that the number of cloudy grid cells increases with higher aerosol concentration, implying a domain-wide increase in cloud droplet concentration and cloud fraction. However, regions with very high LWP values become less pronounced as aerosol concentration increases. This occurs because higher aerosol concentrations lead to the same total liquid water content being partitioned into a larger number of cloud droplets. Consequently, the LWP distribution becomes wider, but with lower peak values. In the A-65 simulation, the cloud pattern resembles the small-domain footprint of a larger open-cell cloud structure, with localized regions of very high LWP values across the domain and smaller clouds in comparison to simulations with higher aerosol concentrations, which may indicate more closed-cell structures. As aerosol concentration increases, LWP values become more broadly distributed throughout the domain, encompassing a larger number of grid cells.

**Figure 2** Snapshots of LWP (g  $m^{-2}$ ) at the end of the  $3^{rd}$  hour of EXP-1 a) A-65, b) A-100, c) A-500, d) A-1000, e) A-10000, and also for BRATIO f) BR-65, g) BR-100, h) BR-500, i) BR-1000, and j) BR-10000.

Figure 3 shows the distribution of cloud cluster effective size for EXP-1, obtained using an object-based classification method using the Connected Component Labelling (CCL) algorithm from Python's

scipy.ndimage module. Cloudy regions are identified by applying a threshold of LWP > 5 g/m<sup>2</sup>, then the algorithm scans the binary field and assigns a unique label to each connected set, using a 4-neighbour connectivity structure. The effective size of each cloud cluster is then calculated from the area covered by connected grid points, and the maximum cluster size can be as large as the simulated domain size of 6.4 km.

Figure 3: Distribution of Cloud cluster effective diameter, in m, for the AERO simulations.

Figure 3 shows that for an aerosol concentration of 65 cm<sup>-3</sup>, the number of clouds with a cluster size less than 1 km is the largest. In contrast, very few clouds exceed cluster size of 1 km. The simulation with aerosol concentration 100 cm<sup>-3</sup> shows the same behaviour but with fewer smaller clouds and sees the appearance of a few cloud structures with a cluster size of around 5 km. These smaller-sized clouds can be considered as small scale equivalent of an open cell structure and are precipitating in nature (Fig.1d). However, on further increase in aerosol concentration, the number of clouds with maximum cluster size equal to domain size of 6.4 km increases. This increase in cluster size can be interpreted as a small-scale equivalent of closed-cell, non-precipitating clouds due to a smaller domain size. The number of clouds with maximum effective cluster size equal to domain size of 6.4 km for 500, 1000, and 10000 aerosol number concentration for the last 2 hours are 19, 23, and 20 clusters respectively, meaning that most of the time domain is covered with large cloud cluster.

Figure 4 presents the vertical profiles averaged horizontally and over the last two hours of simulation for cloud liquid water content, total water content, cloud droplet number concentration, updraft velocity, and vertical velocity variance for the AERO simulations. An enhancement in cloud liquid water content is evident as the aerosol number concentration increases from 65 to 100 cm<sup>-3</sup> (Fig. 4a), which also corresponds to an increase in cloud droplet number (CDNC), vertical velocity variance, and mean updraft velocity (Fig. 4b–d). The enhanced vertical velocity variance under higher aerosol conditions can be attributed to enhanced radiative cooling near the cloud top, which induces stronger turbulent circulations via buoyancy reversal mechanisms. However, with further increases in aerosol number concentration

beyond 100 cm<sup>-3</sup>, a reduction in cloud liquid water content is observed. This decline is likely due to more evaporation of smaller-sized droplets formed under highly polluted conditions with high CDNC (Fig. 4c). Despite the continued increase in aerosol concentrations, not all aerosols are activated into cloud droplets. This behaviour reflects the transitioning into an updraft-limited regime, wherein the relatively weak updrafts characteristic of marine stratocumulus environments is insufficient to activate the abundant available aerosol particles (Fig. 4d), as described in Reutter et al. (2009). Furthermore, the vertical profiles show that simulations with higher aerosol concentrations are associated with elevated inversion heights, potentially resulting from enhanced entrainment and mixing between the cloud layer and the overlying free troposphere.

**Figure 4** Vertical profiles averaged horizontally and over the last two hours of simulation for (a) cloud water content (g kg<sup>-1</sup>), (b) total water mixing ratio (g kg<sup>-1</sup>), (c) cloud droplet number concentration (cm<sup>-3</sup>), (d) updraft velocity (m s<sup>-1</sup>), and (e) vertical velocity variance (m<sup>2</sup> s<sup>-2</sup>) for AERO simulations.

#### 3.2 Transition from updraft-limited to aerosol-limited regime

The strength of updrafts within an air parcel significantly influences aerosol activation into cloud droplets, as expected from the theory governing droplet nucleation dynamics (Pruppacher and Klett, 2012) and from the activation parametrization used in MONC. BRATIO was performed with enhanced latent heat fluxes to investigate the role of thermodynamic forcing on cloud susceptibility, using a reduced Bowen ratio of 0.06. This setup was designed to increase convective vigour and updraft intensity.

Figure 5 presents the time series of horizontally averaged cloud macrophysical properties, including LWP, cloud base and top heights, and precipitation for varying aerosol concentrations under the modified Bowen ratio. The qualitative trends remain broadly consistent with those observed in the AERO simulations: higher aerosol concentrations result in suppressed precipitation and reduced LWP, indicative of a shift in microphysical processes. However, the change in surface flux conditions introduces discernible differences in the spatial structure of the cloud field, as shown in Fig. 2i–k. BRATIO

227228

229230

235236

242243

246247

simulations exhibit larger contiguous regions of high LWP, yet display a net reduction in cloud fraction, suggesting increased heterogeneity in cloud cover and an expansion of cloud-free areas.

**Figure 5.** Horizontally averaged time series evolution of a) liquid water path, b) cloud base height, c) cloud top height, and d) precipitation rates for BRATIO simulations with increasing aerosol concentrations (BR-65 to BR-10000 cm<sup>-3</sup>).

To investigate the contrasting microphysical and macrophysical cloud responses under different aerosol loadings and thermodynamic environments, we now compare simulations with aerosol number concentrations of 65 and 1000 cm<sup>-3</sup>, corresponding to pristine and polluted conditions, respectively, for the AERO and BRATIO cases. Figure 6 shows the vertical profiles averaged over the horizontal and the last two hours of key cloud dynamical and microphysical properties. As shown in Fig. 6d, the enhancement of latent heat flux achieved by lowering the Bowen ratio leads to a noticeable strengthening in the updraft velocity, indicative of enhanced convective activity. This is further supported by the increased vertical velocity variance, shown in Fig. 6e, which is often associated with enhanced turbulent mixing and more vigorous convection. The observed intensification of updraft velocity is expected to promote the activation of a larger fraction of aerosols into cloud droplets due to stronger vertical lifting and supersaturation generation. However, Fig. 6c does not show an apparent increase in cloud droplet number concentration with enhanced latent heating. This apparent discrepancy may arise from spatial and temporal variability, and the differences in droplet activation are instead more effectively captured through domain- and time-averaged quantities discussed in the following sections. Furthermore, the total water content profiles (Fig. 6b) also show increased inversion heights in the BRATIO simulations. This suggests greater entrainment and mixing with the overlying free tropospheric air, consistent with enhanced convective vigour and turbulence in these cases.

**Figure 6** Vertical profiles averaged horizontally and over the last two hours of simulation for (a) cloud water content (g kg $^{-1}$ ), (b) total water mixing ratio (g kg $^{-1}$ ), (c) cloud droplet number concentration (cm $^{-3}$ ), (d) updraft velocity (m s $^{-1}$ ), and (e) vertical velocity variance (m $^2$  s $^{-2}$ ). A-65 and BR-65 represent pristine conditions with Bowen ratios of 0.17 and 0.06, respectively, while A-1000 (violet) and BR-1000 (magenta) represent polluted conditions.

To evaluate the ability of MONC, coupled with the double-moment microphysics CASIM model, to accurately capture the updraft-limited regime and the transition towards aerosol-limited behaviour in marine stratocumulus clouds, Fig. 7 presents the dependence of cloud properties to aerosol perturbations for both experiments: AERO (black lines) and BRATIO (blue lines). The susceptibility parameter  $\beta$  is defined as  $\frac{\partial \ln N_d}{\partial \ln N_a}$ , is equal to 1 for both experiments at the lowest aerosol concentration of 65 to 100 cm<sup>-3</sup>, typically showing the aerosol-limited regime. While with an increase in concentration from 100 to 500, 500 to 1000, and 1000 to 10000 cm<sup>-3</sup>, the  $\beta$  values are 0.78, 0.69, and 0.2, respectively, for AERO, whereas BRATIO shows higher  $\beta$  values of 0.84, 0.71, and 0.34, respectively. It can also be noted from the susceptibility results that with an increase in updraft velocity,  $\beta$  values increase for higher aerosol concentration, and an updraft-limited regime is shifting towards aerosol-limited.

**Figure 7:** Dependence of (a) mean cloud droplet number concentration, in cm<sup>-3</sup>, and (b) liquid water path (LWP, in g m<sup>-2</sup>) to changes in aerosol number concentration under two experiments: AERO (black line) and the BRATIO (blue line).

The LWP susceptibility, defined as  $\beta_{lwp} = \frac{\partial LWP}{\partial lnN_a}$ , shown in Fig. 7b, aligns well with the trends observed in the earlier time series analysis. Specifically, LWP increases from the pristine to moderately polluted cases (65 to 100 cm<sup>-3</sup>), followed by a decline at higher aerosol concentrations. This decline is due to enhanced evaporation of smaller droplets that form in highly polluted environments, as seen in diagnosed evaporative flux (not shown). A similar response is evident in the simulations with stronger updrafts (i.e., BRATIO), where aerosol activation increases as expected. This is corroborated by the blue line in Fig. 7a, where cloud droplet numbers increase more substantially with aerosol loading under enhanced updraft conditions. Taken together, the susceptibility analysis shows that MONC not only captures the expected aerosol-to-updraft limited transition but also responds realistically to dynamical variability, reinforcing its suitability for studying cloud–aerosol interactions.

**Figure 8**. Density scatterplots of cloud droplet number concentration (cm<sup>-3</sup>) and updraft velocity (m s<sup>-1</sup>) at cloud base (lowest level where liquid water contents exceeds 0.001 g kg<sup>-1</sup>) for all simulations. The top row (panels a–e) shows the AERO simulations with increasing aerosol number concentrations (A-series): 65, 100, 500, 1000, and 10000 cm<sup>-3</sup>. The bottom row (panels f–j) shows the BRATIO (BR-series) simulations with enhanced updrafts, using the same corresponding aerosol concentrations.

However, it is also important to note that aerosols and updraft velocity are not independent, blurring the distinction between aerosol-limited and updraft-limited regimes. To explore this coupling further, the relationship between updraft velocity and cloud droplet number concentration is examined using two-dimensional density plots at cloud base (defined as the level where the cloud liquid water content first

exceeds 0.001 g kg<sup>-1</sup>), as shown in Fig. 8. In AERO simulations, maximum density is located at low updraft velocities and low droplet number concentrations in all cases. A positive correlation between updraft velocity and droplet number is evident, with a distinct branch of enhanced density at higher updrafts. Additionally, a second branch is observed, showing higher droplet concentration at low updraft velocities. These branches are primarily due to variable cloud base heights and are discussed in Appendix B. Notably, the cloud droplet number concentration saturates at values significantly lower than the initial aerosol concentrations prescribed in each simulation. The most pronounced linear relationship between updraft and droplet number is observed in Fig. 8e corresponding to an aerosol number concentration of 10000 cm<sup>-3</sup>. In this case, the maximum cloud droplet number concentration reaches approximately 4000 cm<sup>-3</sup> or about 40 % of the available initial aerosol number concentration. This behaviour is consistent with the activation parameterization in the CASIM microphysics scheme (Abdul-Razzak and Ghan, 2000) wherein the fraction of activated aerosols is a function of both the mode radius and the updraft velocity.

A similar behaviour is apparent in BRATIO simulations (Fig. 8f-j), though the density distribution is broader, indicating a higher droplet number for comparable updrafts. A slight increase in the maximum updraft velocity is also evident as observed in Fig. 8j. An additional feature in both sets of simulations is the occurrence of regions with low updraft velocities and relatively high droplet concentrations, which is due to clouds with very high cloud bases that have typically low updraft velocity. Similarly, increased aerosol concentration of 500 and 1000 cm<sup>-3</sup> with higher cloud bases shows the maximum velocity peak for cloud base 400-600 m, and a secondary branch for higher cloud bases (Figure B1).

**Figure 9**. As Figure 8, for the whole cloudy region (all grid points where liquid water content is larger than  $0.001 \text{ g kg}^{-1}$ ).

To further analyse cloud behaviour beyond the cloud base, Fig. 9 shows the overall relationship between updraft velocity and cloud droplet number concentration throughout the cloudy region for AERO and BRATIO. Both sets of experiments exhibit a maximum density at very low updraft velocities (approximately  $\approx 0.1 \text{ m s}^{-1}$ ), with a corresponding cloud droplet number concentration representing roughly 20% of the aerosol number.

In summary, a positive correlation between updraft velocity and cloud droplet number concentration is evident in both Fig. 8 and 9, although density is lower at the highest updraft velocities. A secondary branch is observed in all simulations except A-10000, characterized by high cloud droplet number concentrations at low updraft velocities, even across the entire cloudy region. The influence of enhanced updrafts is clearly reflected in the BRATIO, which exhibits higher maximum updraft velocities and broader density regions. This highlights the role of strengthened updrafts in promoting increased cloud

droplet formation, as previously discussed. Results also suggest that a high-updraft regime can eventually

transition to an aerosol-limited regime, as activation is easier with stronger updrafts.

### 4. Conclusions

This study analyzed the aerosol/updraft-limited regimes in the MONC large eddy simulation model coupled with the two-moment CASIM aerosol-interacting cloud microphysics scheme. The study focused on marine stratocumulus clouds, given their importance in the Earth's energy budget. Simulations were performed from a well-established intercomparison study for marine stratocumulus clouds (Ackermann et al., 2009) field campaign, but with varying aerosol number concentrations of 65, 100, 500, 1000, and 10000 cm<sup>-3</sup> in a first set of simulations denoted AERO. Marine stratocumulus have low updraft velocities; thus, for a very high aerosol concentration, this corresponds to a typically updraft-limited regime. A second set of simulations, denoted BRATIO, performed with enhanced updraft velocity in the stratocumulus clouds, aimed to understand the impact of increased updraft on the transition from the updraft-limited regime. It was achieved by enhancing the surface latent heat flux value to 266.6 W m<sup>-2</sup>, corresponding to a Bowen ratio of 0.06, down from 0.17 in the first set of experiments.

The results show that at low-to-moderate aerosol concentrations (65–100 cm<sup>-3</sup>), an increase in aerosol number enhances cloud droplet number concentration, LWP, and cloud top height, while suppressing precipitation. However, beyond this range, additional aerosol loading leads to a decline in LWP due to entrainment—evaporation feedback. Moreover, the cloud structure shows signs of changing from open to closed cells due to increased cloud droplet numbers leading to large clouds covering the whole of the domain for higher aerosol concentrations. It can be concluded from the results that the weak updraft velocity in stratocumulus clouds restricts the activation of all of the aerosols into cloud droplets, thereby limiting further increases in cloud droplet number, thus well capturing the transition from aerosol limited to updraft-limited regime and flattening the susceptibility curve. The results from the BRATIO

simulations lead to a noticeable strengthening in the updraft velocity, indicative of enhanced convective activity. This is further supported by the increased magnitude of vertical velocity variance, which is often associated with enhanced turbulent mixing and more vigorous convection. As a result, aerosol activation becomes more efficient, leading to higher cloud droplet number concentrations, consistent with the supersaturation dynamics predicted by parcel model theory. Furthermore, the higher  $\beta$  values at high aerosol concentrations indicate that stronger updrafts can shift an updraft-limited regime back towards an aerosol-limited. The findings from the above study showed that both updrafts and aerosols play a crucial role in changing the microphysical and dynamical properties of the stratocumulus clouds, and also highlight the necessity of jointly considering thermodynamic forcing and aerosol variability when evaluating cloud susceptibility and radiative effects in marine environments.

# Appendix A

The cloud top entrainment rate from simulations is computed by

$$E = \frac{dz_i}{dt} + Dz_i$$
 A1

where  $z_i$  is the minimum height of the total water gradient, and D is the divergence.

**Figure A1.** Entrainment velocity (cm s $^{-1}$ ) variation across the AERO for aerosol concentration varying from 65 to  $10^4$  cm $^{-3}$ .

Figure A1 shows the variation of entrainment rate for AERO updraft-limited regime simulations. The entrainment rate also follows the opposite trend observed in LWP, which is shown in Fig. 1, with lower values for A-100 and then increasing with an increase in aerosol concentration, depicting greater evaporation of smaller cloud droplets.

372373

**Figure A2.** Entrainment velocity (cm s $^{-1}$ ) comparison of AERO experiments with BRATIO for aerosol concentrations of 65 and  $10^3$  cm $^{-3}$ .

Moreover, in the BRATIO cases, a clear increment in entrainment rate is observed, as shown in Fig. A2.

The higher entrainment of environmental air leads to more evaporation of cloud droplets and, in turn,

lowers the LWP value as discussed in the main text.

# 375 Appendix B

**Figure B1.** Density scatterplots of cloud droplet number concentration (cm $^{-3}$ ) and updraft velocity (m s $^{-1}$ ) at different cloud bases. The rows show increasing cloud bases, for clouds with bases between (from top to bottom) 200-400 m, 400-600 m, 600-800 m, 800-1000 m. Columns correspond to AERO simulations with initial aerosol concentrations of (from left to right) 65, 500 and 1000 cm $^{-3}$ .

In order to clearly comprehend the impact of cloud bases on the second branch, the distributions of cloud droplet number concentrations and updraft velocities are divided based on different cloud base height groups. Figure B1 presents the variation of density distributions for different cloud bases of AERO simulations for aerosol concentrations of 65, 500, and 1000 cm<sup>-3</sup>. The reason for the second branch with low updraft velocity and high cloud droplet number (as shown previously in Fig. 8) can be observed clearly. For the A-65 case with a lower cloud base (shown in Fig. 1b), high updraft velocity with maximum density is found for cloud base heights of 200–400 m (Fig. B a). As the cloud base height

- increases from 600-1000 m, the updraft velocity weakens, and the secondary branch becomes more
- apparent at these higher cloud bases.

# 390 Code Availability

- MONC and CASIM are managed on UK Met Office repository code.metoffice.gov.uk, which requires
- registration.

### 393 Data Availability

Output of simulations will be made available on Zenodo.

#### 395 Authors contribution

- GD, OB and NB designed the experiments and GD carried them out. GD prepared the manuscript with
- contributions from all co-authors.

#### 398 Competing interests

The authors declare that they have no conflict of interest.

#### 400 Acknowledgements

- The MONC source code has been provided by the UK Met Office, and simulations were conducted on
- the IPSL-MESO Spirit Cluster.

### 403 Financial support

- The research has been funded by the European Union under the grant agreement no 101137639:
- CleanCloud.

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
