# Peer review of "Interplay between aerosol and updraft velocity in Large Eddy"

_EGUsphere, 2025_

## Referee Comment (RC1)

**Review of „Interplay between aerosol and updraft velocity in Large Eddy Simulations of marine stratocumulus clouds" by Dogra, G., Boucher, O., and Bellouin, N. (egusphere-2025-5711)**

This manuscript uses LES simulations, initialized with observational data from the DYCOMS-II field campaign, with a bulk microphysics scheme to simulate and study stratocumulus clouds under varying aerosol burdens and latent heat fluxes. The increased latent heat flux is chosen to produce stronger updrafts, this response is confirmed throughout the manuscript. Therefore, the parameters varied to study the response of the stratocumulus-topped boundary layer are aerosol number concentration and updraft velocity. The authors use the obtained data to gain information about the behavior of more realistic cloud systems in the so-called aerosol-limited and updraft-limited regimes of condensating cloud droplets and to demonstrate the ability of their model to capture both regimes, as well as the transition from one into the other.

While this study provides interesting proof-of-concept data and some key points of the responses of the cloud system, I do not think that the work is developed enough to be published. The additional efforts required to elaborate on the simulations, as well as model, physical, and numerical responses in my opinion exceed a major revision process. Inadmissible reasoning is found in crucial places. I suggest re-submission after narrowing down the reasons for observed effects and clarification of the overall intention and result of the paper.

**Main comments:**

*Aerosol number over time.* As the aerosol concentration will decrease due to collision and coalescence (and maybe deposition if included in the simulations), a time series of the mean aerosol concentration (and cloud droplet concentration) over time would be very informative to interpret the results. It was also not clear to me if there is an aerosol source.

*Microphysics model.* The work presented relies heavily on the representation of activation at cloud base. The authors should add a chapter critically evaluating the employed microphysics scheme and elaborating on the representation of supersaturation in it. While it is substantial to investigate the effects and behavior of the, compared to more sophisticated bin or Lagrangian schemes, less accurate but more efficient bulk schemes in aerosol-cloud-interactions, it is necessary to position the used scheme in the variety of used parameterizations.

*Figure 8.* The plots in this figure are very interesting, but lack sufficient discussion.
- The authors should mention how they define a cloud droplet.
- It is not clear which time points are used for the data in this plot. This should be mentioned.
- Especially for decreasing aerosol concentration in the domain, it might be useful to plot the activated fraction of aerosol instead of the absolute activated cloud droplet number. I would like to suggest to the authors to see if this gives less noisy plots, which might turn out not to be the case.
- I cannot follow the argumentation of lines 300 to 303. The reference given mentions varying activated number fractions for similar aerosol conditions and lower updrafts (but different thermodynamics). Moreover, roughly 40% activated number fraction is a barrier

in all presented plots in Figure 8. I wonder if this is a drawback of the microphysics parametrization or some other effect. I would like to request a clarification of this distinct feature in the plots. I think it is not very likely to be realistic, as, e.g., 65 aerosols per cubic centimeter should have a significantly higher activated fraction than 40% for a mean radius of 0.06µm and 2m/s updraft velocity.

- The reasoning in lines 306 to 308 is not clear to me. I cannot follow why high cloud bases produce such high activated number fractions for low updraft velocities, notably the only activated number fractions above 40%. While the fact that this branch stems from higher cloud bases is indeed shown in the appendix, the reasoning for it is invalid. It is absolutely necessary to rigorously track down what is causing this, as there are multiple possible reasons with varying implications for this work. Processes that should be investigated include convergence from advection schemes that are not divergence-free, accumulation of cloud droplets in an updraft by additional sedimentation from above, spurious supersaturations (https://doi.org/10.1175/1520-0493(1996)124<1034:TSPOCE>2.0.CO;2), errors in supersaturation calculation, and spatial and temporal resolution of the simulation.

- Particularly, elaborating on Figure 8 would be informative with respect to deviation from parcel theory. A solid line in the plot showing the activated aerosol number of an adiabatic parcel using the same microphysics parametrization could be shown. I suppose this might be the line of high density approaching roughly 40% for stronger updrafts. From this, the other features observable in the plots might be discussed and explained thoroughly. This would also enable a discussion of the deviations of more realistic clouds from parcel-theory-driven aerosol- and updraft-limited regime discussions.

**Minor comments:**

*Model response or physical response?* In the light of the authors' aim to investigate their models capability of representing the aerosol- and updraft-limited regimes, a more critical evaluation of the microphysics scheme and its strengths and weaknesses in these simulations would be appropriate. This is partly covered in the main comments.

*LWP distribution.* In Lines 157 to 158, it is stated that a larger number of cloud droplets leads to a wider LWP distribution with lower peak values. This statement has not been reasoned. The word "Consequently" is not adequate and elaboration is necessary. I also am not convinced that this is actually apparent, since the effect which is significant from Fig. 2 is the transition to closed cells due to precipitation suppression, but the broader LWP distribution is, if evident, more likely to be caused by vanishing precipitation-induced cold pools which cease to create strong updrafts. Aerosol increases beyond that lead to a decrease in LWP due to evaporation-entrainment feedback, as indicated by the authors in lines 197-198. The total liquid water content does not stay the same.

*Stratocumulus or cumulus?* The small patches of high liquid water path in Fig. 1 and the behavior described in line 224 indicate beginning cumuli-form convection behavior. A representative x-z slice showing cloud water content through the domain could give the reader a better idea of the clouds in the simulations.

**Technical comments:**

Consistent layouts should be used for Figs. 1 and 5, since they show the same thing for two different sets of simulations.

Figure 4c: It could be useful here to average only over cloudy grid boxes to give more realistic values of cloud droplet numbers.

Figs. 1 and 5: The definition of cloud base height and cloud top is not clear.

Ll. 26-28: The second part of the sentence might be misleading, as the limited vertical extent is not the crucial factor, but the combination of low cloud base and limited vertical extend. I suggest reformulation to „Being low-level clouds with limited vertical extent, their effect on longwave radiation at the top of the atmosphere is minimal.".

Ll. 33: The word „Thus" implies that the aforementioned is the reason for the following statement. This is not true in this case, as the susceptibility of marine stratocumulus is neither due to supersaturations ranging between 0.1% to 2%, nor from being warm clouds (but rather due to limited aerosol concentration and LWP).

Line 49: The word „adiabatically" is superfluous here, since no parcel in the atmosphere is lifted truly adiabatically.

Line 116: The words „varied across four cases" are followed by five numbers. This might be a typo.

Line 150: It seemed to me that there is no precipitation for aerosol concentrations above 100 per cubic centimeter. There cannot be any subsequent precipitation suppression then. While the cloud base plots do show an increase for increasing aerosol concentrations, the cloud tops seem to stay at the same level. The increased cloud base might be due to a drying boundary layer. This is the result from increased mixing with dry above-cloud air for larger aerosol concentration, which was indicated by the authors.

Line 171: A consistent threshold to identify cloudy regions should be used to increase accessibility and consistency of the manuscript. In Fig. 2, it is 2 g/m2, while in line 171, it is 5 g/m2.

Line 181: There also seem to be clusters at 2 and 3km.

Line 181: „These" should be „The", otherwise the 5km clouds are regarded to as smaller-sized.

Line 361: „$z_i$ is the minimum height of the total water gradient" - This statement is not clear. The total water content has a gradient everywhere. Maybe the height of the maximum total water gradient was meant.